# Learning Deep Parsimonious Representations

**Renjie Liao[1], Alexander Schwing[2], Richard S. Zemel[1,3], Raquel Urtasun[1]**
University of Toronto[1]
University of Illinois at Urbana-Champaign[2]
Canadian Institute for Advanced Research[3]
{rjliao, zemel, urtasun}@cs.toronto.edu, aschwing@illinois.edu

## Abstract

In this paper we aim at facilitating generalization for deep networks while supporting interpretability of the learned representations. Towards this goal, we propose a clustering based regularization that encourages parsimonious representations. Our k-means style objective is easy to optimize and flexible, supporting various forms of clustering, such as sample clustering, spatial clustering, as well as co-clustering. We demonstrate the effectiveness of our approach on the tasks of unsupervised learning, classification, fine grained categorization, and zero-shot learning.

## 1 Introduction

In recent years, deep neural networks have been shown to perform extremely well on a variety of tasks including classification [21], semantic segmentation [13], machine translation [27] and speech recognition [16]. This has led to their adoption across many areas such as computer vision, natural language processing and robotics [16, 21, 22, 27]. Three major advances are responsible for the recent success of neural networks: the increase in available computational resources, access to large scale data sets, and several algorithmic improvements.

Many of these algorithmic advances are related to regularization, which is key to prevent overfitting and improve generalization of the learned classifier, as the current trend is to increase the capacity of neural nets. For example, batch normalization [18] is used to normalize intermediate representations which can be interpreted as imposing constraints. In contrast, dropout [26] removes a fraction of the learned representations at random to prevent co-adaptation. Learning of de-correlated activations [6] shares a similar idea since it explicitly discourages correlation between the units.

In this paper we propose a new type of regularization that encourages the network representations to form clusters. As a consequence, the learned feature space is compactly representable, facilitating generalization. Furthermore, clustering supports interpretability of the learned representations. We formulate our regularization with a k-means style objective which is easy to optimize, and investigate different types of clusterings, including sample clustering, spatial clustering, and co-clustering.

We demonstrate the generalization performance of our proposed method in several settings: autoencoders trained on the MNIST dataset [23], classification on CIFAR10 and CIFAR100 [20], as well as fine-grained classification and zero-shot learning on the CUB-200-2011 dataset [34]. We show that our approach leads to significant wins in all these scenarios. In addition, we are able to demonstrate on the CUB-200-2011 dataset that the network representation captures meaningful part representations even though it is not explicitly trained to do so.

## 2 Related Work

Standard neural network regularization involves penalties on the weights based on the norm of the parameters [29, 30]. Also popular are regularization methods applied to intermediate representations,

such as Dropout [26], Drop-Connect [32], Maxout [10] and DeCov [6]. These approaches share the aim of preventing the activations in the network to be correlated. Our work can be seen as a different form of regularization, where we encourage parsimonious representations.

A variety of approaches have applied clustering to the parameters of the neural network with the aim of compressing the network. Compression rates of more than an order of magnitude were demonstrated in [11] without sacrificing accuracy. In the same spirit hash functions were exploited in [5]. Early approaches to compression include biased weight decay [12] and [14, 24], which prunes the network based on the Hessian of the loss function.

Recently, various combinations of clustering with representation learning have been proposed. We categorize them broadly into two areas: (i) work that applies clustering after having learned a representation, and (ii) approaches that jointly optimize the learning and clustering objectives. [4] combines deep belief networks (DBN) with non-parametric maximum-margin clustering in a post-hoc manner: A DBN is trained layer-wise to obtain an intermediate representation of the data; non-parametric maximum-margin clustering is then applied to the data representation. Another line of work utilizes an embedding of the deep network, which can be based on annotated data [15], or from a learned unsupervised method such as a stacked auto-encoder [28]. In these approaches, the network is trained to approximate the embedding, and subsequently either k-means or spectral clustering is performed to partition the space. An alternative is to use non-negative matrix factorization, which represents a given data matrix as the product of components [31]. This deep non-negative matrix factorization is trained using the reconstruction loss rather than a clustering objective. Nonetheless, it was shown that factors lower in the hierarchy have superior clustering performance on low-level concepts while factors later in the hierarchy cluster high-level concepts. The aforementioned approaches differ from our proposed technique, since we aim at jointly learning a representation that is parsimonious via a clustering regularization.

Also related are approaches that utilize sparse coding. Wang et al. [33] unrolls the iterations forming the sparse codes and optimizes end-to-end the involved parameters using a clustering objective as loss function [33]. The proposed framework is further augmented by clustering objectives applied to intermediate representations, which act as feature regularization within the unrolled optimization. They found that features lower in the unrolled hierarchy cluster low-level concepts, while features later in the hierarchy capture high-level concepts. Our method differs in that we use convolutional neural networks rather than unrolling a sparse coding optimization.

In the context of unsupervised clustering [35] exploited agglomerative clustering as a regularizer; this approach was formulated as a recurrent network. In contrast we employ a k-means like clustering objective which simplifies the optimization significantly and does not require a recurrent procedure. Furthermore, we investigate both unsupervised and supervised learning.

## 3 Learning Deep Parsimonious Representations

In this section, we introduce our new clustering based regularization which not only encourages the neural network to learn more compact representations, but also enables interpretability of the neural network. We first show that by exploiting different unfoldings of the representation tensor, we obtain multiple types of clusterings, each possessing different properties. We then devise an efficient online update to jointly learn the clustering with the parameters of the neural network.

### 3.1 Clustering of Representations

We first introduce some notation. We refer to $[K]$ as the set of $K$ positive integers, *i.e.*, $[K] = \{1, 2, ..., K\}$. We use $\mathcal{S} \backslash \mathcal{A}$ to denote the set $\mathcal{S}$ with elements from the set $\mathcal{A}$ removed. A tensor is a multilinear map over a set of vector spaces. In tensor terminology, *n-mode* vectors of a $D$-order tensor $\mathbf{Y} \in \mathbb{R}^{I_1 \times I_2 \times \cdots \times I_D}$ are $I_n$-dimensional vectors obtained from $\mathbf{Y}$ by varying the index in $I_n$-dimension, while keeping all other indices fixed. An *n-mode* matrix unfolding of a tensor is a matrix which has all *n-mode* vectors as its columns [7]. Formally we use the operator $T^{\{I_n\} \times \{I_j | j \in [D] \backslash n\}}$ to denote the *n-mode* matrix unfolding, which returns a matrix of size $I_n \times \prod_{j \in [D] \backslash n} I_j$. Similarly, we definee $T^{\{I_i, I_j\} \times \{I_k | k \in [D] \backslash \{i, j\}\}}$ to be an $(i, j)$-*mode* matrix unfolding operator. In this case a column vector is a concatenation of one *i-mode* vector and one *j-mode* vector. We denote the $m$-th row vector of a matrix $\mathbf{X}$ as $\mathbf{X}_m$.

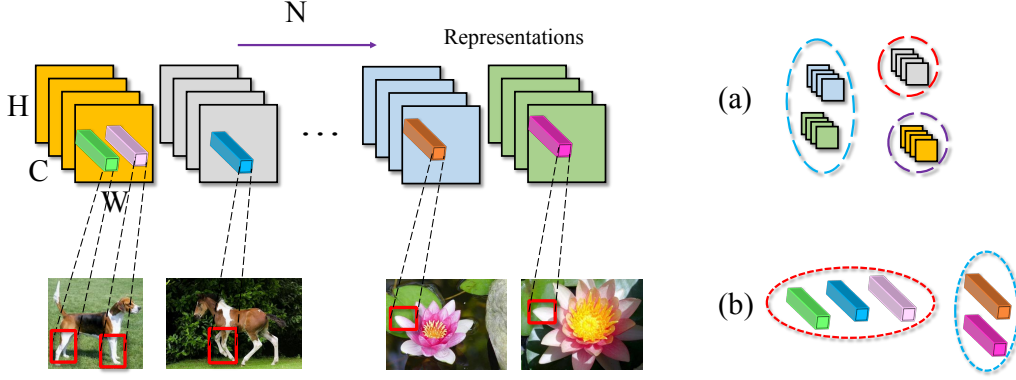

Figure 1: (A) Sample clustering and (B) spatial clustering. Samples, pixels, and channels are visualized as multi-channel maps, cubes, and maps in depth respectively. The receptive fields in the input image are denoted as red boxes.

In this paper we assume the representation of one layer within a neural network to be a 4-D tensor $\mathbf{Y} \in \mathbb{R}^{N \times C \times H \times W}$, where $N$, $C$, $H$ and $W$ are the number of samples within a mini-batch, the number of hidden units, the height and width of the representation respectively. Note that $C$, $H$ and $W$ can vary between layers, and in the case of a fully connected layer, the dimensions along height and width become a singleton and the tensor degenerates to a matrix.

Let $\mathcal{L}$ be the loss function of a neural network. In addition, we refer to the clustering regularization of a single layer via $\mathcal{R}$. The final objective is $\mathcal{L} + \lambda \mathcal{R}$, where $\lambda$ adjusts the importance of the clustering regularization. Note that we can add a regularization term for any subset of layers, but we focus on a single layer for notational simplicity. In what follows, we show three different types of clustering, each possessing different properties. In our framework any variant can be applied to any layer.

**(A) Sample Clustering:** We first investigate clustering along the sample dimension. Since the cluster assignments of different layers are not linked, each layer is free to cluster examples in a different way. For example, in a ConvNet, bottom layer representations may focus on low-level visual cues, such as color and edges, while top layer features may focus on high-level attributes which have a more semantic meaning. We refer the reader to Fig. 1 (a) for an illustration. In particular, given the representation tensor $\mathbf{Y}$, we first unfold it into a matrix $T^{\{N\} \times \{H,W,C\}}(\mathbf{Y}) \in \mathbb{R}^{N \times HWC}$. We then encourage the samples to cluster as follows:

$$\mathcal{R}_{sample}(\mathbf{Y}, \mu) = \frac{1}{2NCHW} \sum_{n=1}^{N} \left\| T^{\{N\} \times \{H,W,C\}}(\mathbf{Y})_n - \mu_{z_n} \right\|^2, \qquad (1)$$

where $\mu$ is a matrix of size $K \times HWC$ encoding all cluster centers, with $K$ the total number of clusters. $z_n \in [K]$ is a discrete latent variable corresponding to the $n$-th sample. It indicates which cluster this sample belongs to. Note that for a fully connected layer, the formulation is the same except that $T^{\{N\} \times \{H,W,C\}}(\mathbf{Y})_n$ and $\mu_{z_n}$ are $C$-sized vectors since $H = W = 1$ in this case.

**(B) Spatial Clustering:** The representation of one sample can be regarded as a $C$-channel "image." Each spatial location within that "image" can be thought of as a "pixel," and is a vector of size $C$ (shown as a colored bar in Fig. 1). For a ConvNet, every "pixel" has a corresponding receptive field covering a local region in the input image. Therefore, by clustering "pixels" of all images during learning, we expect to model local parts shared by multiple objects or scenes. To achieve this, we adopt the unfolding operator $T^{\{N,H,W\} \times \{C\}}(\mathbf{Y})$ and use

$$\mathcal{R}_{spatial}(\mathbf{Y}, \mu) = \frac{1}{2NCHW} \sum_{i=1}^{NHW} \| T^{\{N,H,W\} \times \{C\}}(\mathbf{Y})_i - \mu_{z_i} \|^2. \qquad (2)$$

Note that although we use the analogy of a "pixel," when using text data a "pixel" may corresponds to words. For spatial clustering the dimension of the matrix $\mu$ is $K \times C$.

**(C) Channel Co-Clustering:** This regularizer groups the channels of different samples directly, thus co-clustering samples and filters. We expect this type of regularization to model re-occurring

---

**Algorithm 1** : Learning Parsimonious Representations

---

1: **Initialization:** Maximum training iteration $R$, batch size $B$, smooth weight $\alpha$, set of clustering layers $\mathcal{S}$ and set of cluster centers $\{\mu_k^0 | k \in [K]\}$, update period $M$
2: **For** iteration $t = 1, 2, ..., R$:
3:     **For** layer $l = 1, 2, ..., L$:
4:         Compute the output representation of layer $l$ as $x$.
5:         **If** $l \in \mathcal{S}$:
6:             Assigning cluster $z_n = \underset{k}{\mathrm{argmin}} \ \|\mathbf{X}_n - \mu_k^{t-1}\|^2, \forall n \in [B]$.
7:             Compute cluster center $\hat{\mu}_k = \frac{1}{|\mathcal{N}_k|} \sum_{n \in \mathcal{N}_k} \mathbf{X}_n$, where $\mathcal{N}_k = [B] \bigcap \{n | z_n = k\}$.
8:             Smooth cluster center $\mu_k^t = \alpha \hat{\mu}_k + (1 - \alpha)\mu_k^{t-1}$
9:         **End**
10:     **End**
11:     Compute the gradients with cluster centers $\mu_k^t$ fixed.
12:     Update weights.
13:     Update drifted cluster centers using Kmeans++ every $M$ iterations.
14: **End**

---

patterns shared not only among different samples but also within each sample. Relying on the unfolding operator $T^{\{N,C\} \times \{H,W\}}(\mathbf{Y})$, we formulate this type of clustering objective as

$$\mathcal{R}_{channel}(\mathbf{Y}, \mu) = \frac{1}{2NCHW} \sum_{i=1}^{NC} \|T^{\{N,C\} \times \{H,W\}}(\mathbf{Y})_i - \mu_{z_i}\|^2. \qquad (3)$$

Note that the dimension of the matrix $\mu$ is $K \times HW$ in this case.

## 3.2 Efficient Online Update

We now derive an efficient online update to jointly learn the weights while clustering the representations of the neural network. In particular, we illustrate the sample clustering case while noting that the other types can be derived easily by applying the corresponding unfolding operator. For ease of notation, we denote the unfolded matrix $T^{\{N\} \times \{H,W,C\}}(\mathbf{Y})$ as $\mathbf{X}$. The gradient of the clustering regularization layer w.r.t. its input representation $\mathbf{X}$ can be expressed as,

$$\frac{\partial \mathcal{R}}{\partial \mathbf{X}_n} = \frac{1}{NCHW} \left[ \mathbf{X}_n - \mu_{z_n} - \frac{1}{Q_{z_n}} \sum_{z_p = z_n, \forall p \in [N]} \left( \mathbf{X}_n - \mu_{z_p} \right) \right], \qquad (4)$$

where $Q_{z_n}$ is the number of samples which belong to the $z_n$-th cluster. This gradient is then backpropagated through the network to obtain the gradient w.r.t. the parameters of the network.

The time and space complexity of the gradient computation of one regularization layer are $\max(\mathcal{O}(KCHW), \mathcal{O}(NCHW))$ and $\mathcal{O}(NCHW)$ respectively. Note that we can cache the centered data $\mathbf{X}_n - \mu_{z_n}$ in the forward pass to speed up the gradient computation.

The overall learning algorithm of our framework is summarized in Alg. 1. In the forward pass, we first compute the representation of the $n$-th sample as $\mathbf{X}_n$ for each layer. We then infer the latent cluster label $z_n$ for each sample based on the distance to the cluster centers $\mu_k^{t-1}$ from the last time step $t - 1$, and assign the sample to the cluster center which has the smallest distance. Once all the cluster assignments are computed, we estimate the cluster centers $\hat{\mu}_k$ based on the new labels of the current batch.

We then combine the estimate based on the current batch with the former cluster center. This is done via an online update. We found an online update together with the random restart strategy to work well in practice, as the learning of the neural network proceeds one mini-batch at a time, and as it is too expensive to recompute the cluster assignment for all data samples in every iteration. Since we trust our current cluster center estimate more than older ones, we smooth the estimation by using an exponential moving average. The cluster center estimate at iteration $t$ is obtained via $\mu_k^t = \alpha \hat{\mu}_k + (1 - \alpha)\mu_k^{t-1}$, where $\alpha$ is a smoothing weight. However, as the representation learned by the neural network may go through drastic changes, especially in the beginning of training, some

| Measurement | Train | Test |
|---|---|---|
| AE | $2.69 \pm 0.12$ | $3.61 \pm 0.13$ |
| AE + Sample-Clustering | $2.73 \pm 0.01$ | $3.50 \pm 0.01$ |

Table 1: Autoencoder Experiments on MNIST. We report the average of mean reconstruction error over 4 trials and the corresponding standard deviation.

| Dataset | CIFAR10 Train | CIFAR10 Test | CIFAR100 Train | CIFAR100 Test |
|---|---|---|---|---|
| Caffe | $94.87 \pm 0.14$ | $76.32 \pm 0.17$ | $68.01 \pm 0.64$ | $46.21 \pm 0.34$ |
| Weight Decay | $95.34 \pm 0.27$ | $76.79 \pm 0.31$ | $69.32 \pm 0.51$ | $46.93 \pm 0.42$ |
| DeCov | $88.78 \pm 0.23$ | $79.72 \pm 0.14$ | $77.92$ | $40.34$ |
| Dropout | $99.10 \pm 0.17$ | $77.45 \pm 0.21$ | $60.77 \pm 0.47$ | $48.70 \pm 0.38$ |
| Sample-Clustering | $89.93 \pm 0.19$ | $\mathbf{81.05} \pm 0.41$ | $63.60 \pm 0.55$ | $\mathbf{50.50} \pm 0.38$ |
| Spatial-Clustering | $90.50 \pm 0.05$ | $\mathbf{81.02} \pm 0.12$ | $64.38 \pm 0.38$ | $\mathbf{50.18} \pm 0.49$ |
| Channel Co-Clustering | $89.26 \pm 0.25$ | $\mathbf{80.65} \pm 0.23$ | $63.42 \pm 1.34$ | $\mathbf{49.80} \pm 0.25$ |

Table 2: CIFAR10 and CIFAR 100 results. For DeCov, no standard deviation is provided for the CIFAR100 results [6]. All our approaches outperform the baselines.

of the cluster centers may quickly be less favored and the number of incoming samples assigned to it will be largely reduced. To overcome this issue, we exploit the Kmeans++ [3] procedure to re-sample the cluster center from the current mini-batch. Specifically, denoting the the distance between sample $\mathbf{X}_n$ and its nearest cluster center as $d_n$, the probability of taking $\mathbf{X}_n$ as the new cluster center is $d_n^2 / \sum_i d_i^2$. After sampling, we replace the old cluster center with the new one and continue the learning process. In practice, at the end of every epoch, we apply the kmeans++ update to cluster centers for which the number of assigned samples is small. See Alg. 1 for an outline of the steps. The overall procedure stabilizes the optimization and also increases the diversity of the cluster centers.

In the backward pass, we fix the latest estimation of the cluster centers $\mu_k^t$ and compute the gradient of loss function and the gradient of the clustering objective based on Eq. (4). Then we back-propagate all the gradients and update the weights.

## 4 Experiments

In this section, we conduct experiments on unsupervised, supervised and zero-shot learning on several datasets. Our implementation based on TensorFlow [9] is publicly available.[1] For initializing the cluster centers before training, we randomly choose them from the representations obtained with the initial network.

### 4.1 Autoencoder on MNIST

We first test our method on the unsupervised learning task of training an autoencoder. Our architecture is identical to [17]. For ease of training we did not tie the weights between the encoder and the decoder. We use the squared $\ell_2$ reconstruction error as the loss function and SGD with momentum. The standard training-test-split is used. We compute the mean reconstruction error over all test images and repeat the experiments 4 times with different random initializations. We compare the baseline model, *i.e.*, a plain autoencoder, with one that employs our sample-clustering regularization on all layers except the top fully connected layer. Sample clustering was chosen since this autoencoder only contains fully connected layers. The number of clusters and the regularization weight $\lambda$ of all layers are set to 100 and $1.0e^{-2}$ respectively. For both models the same learning rate and momentum are used. Our exact parameter choices are detailed in the Appendix. As shown in Table 1, our regularization facilitates generalization as it suffers less from overfitting. Specifically, applying our regularization results in lower test set error despite slightly higher training error. More importantly, the standard deviation of the error is one order of magnitude smaller for both training and testing when applying our regularization. This indicates that our sample-clustering regularization stabilizes the model.

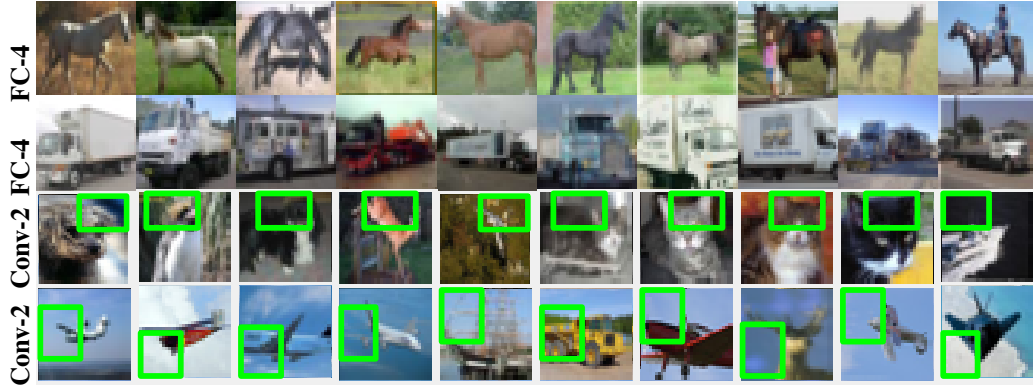

Figure 2: Visualization of clusterings on CIFAR10 dataset. Rows 1, 2 each show examples belonging to a single sample-cluster; rows 3, 4 show regions clustered via spatial clustering.

## 4.2 CIFAR10 and CIFAR100

In this section, we explore the CIFAR10 and CIFAR100 datasets [20]. CIFAR10 consists of 60,000 $32 \times 32$ images assigned to 10 categories, while CIFAR100 differentiates between 100 classes. We use the standard split on both datasets. The quick CIFAR10 architecture of Caffe [19] is used for benchmarking both datasets. It consists of 3 convolutional layers and 1 fully connected layer followed by a softmax layer. The detailed parameters are publicly available on the Caffe [19] website. We report mean accuracy averaged over 4 trials. For fully connected layers we use the sample-clustering objective. For convolutional layers, we provide the results of all three clustering objectives, which we refer to as 'sample-clustering,' 'spatial-clustering,' and 'channel-co-clustering' respectively. We set all hyper-parameters based on cross-validation. Specifically, the number of cluster centers are set to 100 for all layers for both CIFAR10 and CIFAR100. $\lambda$ is set to $1.0e^{-3}$ and $1.0e^{-2}$ for the first two convolutional and the remaining layers respectively in CIFAR10; for CIFAR100, $\lambda$ is set to 10 and 1 for the first convolutional layer and the remaining layers respectively. The smoothness parameter $\alpha$ is set to 0.9 and 0.95 for CIFAR10 and CIFAR100 respectively.

**Generalization:** In Table 2 we compare our framework to some recent regularizers, like DeCov [6], Dropout [26] and the baseline results obtained using Caffe. We again observe that all of our methods achieve better generalization performance.

**Visualization:** To demonstrate the interpretability of our learned network, we visualize sample-clustering and spatial-clustering in Fig. 2, showing the top-10 ranked images and parts per cluster. In the case of sample-clustering, for each cluster we rank all its assigned images based on the distance to the cluster center. We chose to show 2 clusters from the 4th fully connected layer. In the case of spatial-clustering, we rank all "pixels" belonging to one cluster based on the distance to the cluster center. Note that we have one part (*i.e.*, one receptive field region in the input image) for each "pixel." We chose to show 2 clusters from the 2nd convolutional layer. The receptive field of the 2nd convolutional layer is of size $18 \times 18$ in the original $32 \times 32$ sized image. We observe that clusterings of the fully connected layer representations encode high-level semantic meaning. In contrast, clusterings of the convolutional layer representations encode attributes like shape. Note that some parts are uninformative which may be due to the fact that images in CIFAR10 are very small. Additional clusters and visualizations on CIFAR100 are shown in the Appendix.

**Quantitative Evaluation of Parsimonious Representation:** We quantitatively evaluate our learned parsimonious representation on CIFAR100. Since only the image category is provided as ground truth, we investigate sample clustering using the 4th fully connected layer where representations capture semantic meaning. In particular, we apply K-means clustering to the learned representation extracted from the model with and without sample clustering respectively. For both cases, we set the number of clusters to be 100 and control the random seed to be the same. The most frequent class label within one cluster is assigned to all of its members. Then we compute the normalized mutual information (NMI) [25] to measure the clustering accuracy. The average results over 10 runs are shown in Table 3. Our representations achieve significantly better clustering quality

| Method | Baseline | Sample-Clustering |
|--------|----------|-------------------|
| NMI | $0.4122 \pm 0.0012$ | $0.4914 \pm 0.0011$ |

Table 3: Normalized mutual information of sample clustering on CIFAR100.

| Method | Train | Test |
|--------|-------|------|
| DeCAF [8] | - | 58.75 |
| Sample-Clustering | 100.0 | **61.77** |
| Spatial-Clustering | 100.0 | 61.67 |
| Channel Co-Clustering | 100.0 | 61.49 |

Table 4: Classification accuracy on CUB-200-2011.

compared to the baseline which suggests that they are distributed in a more compact way in the feature space.

## 4.3 CUB-200-2011

Next we test our framework on the Caltech-UCSD birds dataset [34] which contains 11,788 images of 200 different categories. We follow the dataset split provided by [34] and the common practice of cropping the image using the ground-truth bounding box annotation of the birds [8, 36]. We use Alex-Net [21] pretrained on ImageNet as the base model and adapt the last layer to fit classification of 200 categories. We resize the image to $227 \times 227$ to fit the input size. We add clusterings to all layers except the softmax-layer. Based on cross-validation, the number of clusters are set to 200 for all layers. For convolutional layers, we set $\lambda$ to $1.0e^{-5}$ for the first (bottom) 2 and use $1.0e^{-4}$ for the remaining ones. For fully connected layers, we set $\lambda$ to $1.0e^{-3}$ and $\alpha$ is equal to $0.5$. We apply Kmeans++ to replace cluster centers with less than 10 assigned samples at the end of every epoch.

**Generalization:** We investigate the impact of our parsimonious representation on generalization performance. We compare with the DeCAF result reported in [8], which used the same network to extract a representation and applied logistic regression on top for fine-tuning. We also fine-tune Alex-Net which uses weight-decay and Dropout, and report the best result we achieved in Table 4. We observe that for the Alex-Net architecture our clustering improves the generalization compare to direct fine-tuning and the DeCAF result. Note that Alex-Net pretrained on ImageNet easily overfits on this dataset as all training accuracies reach 100 percent.

**Visualization:** To visualize the sample-clustering and spatial-clustering we follow the setting employed when evaluating on the CIFAR dataset. For the selected cluster center we show the 10 closest images in Fig. 3. For sample clustering, 2 clusters from the 3rd convolutional layer and the 7th fully connected layer are chosen for visualization. For spatial clustering, 2 clusters from the 2nd and 3rd convolutional layers are chosen for visualization. More clusters are shown in the Appendix. The receptive fields of pixels from the 2nd and 3rd convolutional layers are of sizes $59 \times 59$ and $123 \times 123$ in the resized $227 \times 227$ image. We observe that cluster centers of sample clustering applied to layers lower in the network capture pose and shape information, while cluster centers from top layers model the fine-grained categories of birds. For spatial clustering, cluster centers from different layers capture parts of birds in different scales, like the beak, chest, etc.

## 4.4 Zero-Shot Learning

We also investigate a zero-shot setting on the CUB dataset to see whether our parsimonious representation is applicable to unseen categories. We follow the setting in [1, 2] and use the same split where 100, 50 and 50 classes are used as training, validation and testing (unseen classes). We use a pre-trained Alex-Net as the baseline model and extract 4096-dimension representations from the 7th fully connected (fc) layer. We compare sample-clustering against other recent methods which also report results of using 7th fc feature of Alex-Net. Given these features, we learn the output embedding $W$ via the same unregularized structured SVM as in [1, 2]:

$$\min_{W} \quad \frac{1}{N} \sum_{n=1}^{N} \max_{y \in \mathcal{Y}} \left\{ 0, \Delta(y_n, y) + x_n^\top W \left[ \phi(y) - \phi(y_n) \right] \right\}, \tag{5}$$

where $x_n$ and $y_n$ are the feature and class label of the $n$-th sample and $\Delta$ is the 0-1 loss function.. $\phi$ is the class-attribute matrix provided by the CUB dataset, where each entry is a real-valued score

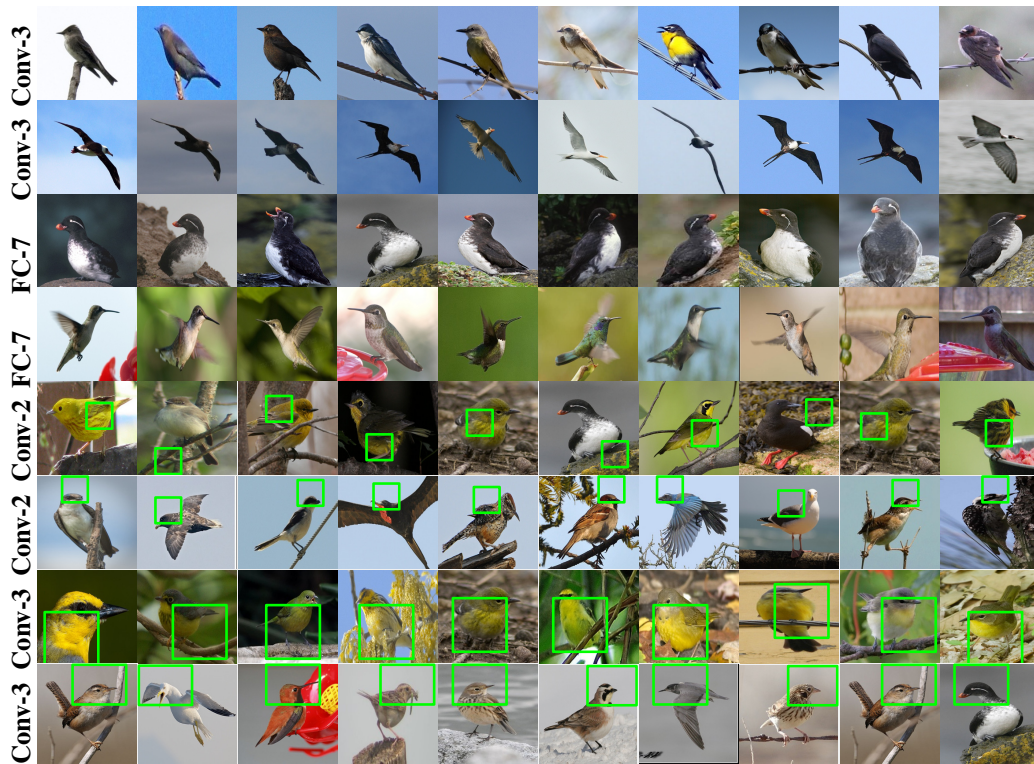

Figure 3: Visualization of sample and pixel clustering on CUB-200-2011 dataset. Row 1-4 and 5-8 show sample and spatial clusters respectively. Receptive fields are truncated to fit images.

| Method | Top1 Accuracy |
|---|---|
| ALE [1] | 26.9 |
| SJE [2] | 40.3 |
| Sample-Clustering | **46.1** |

Table 5: Zero-shot learning on CUB-200-2011.

indicating how likely a human thinks one attribute is present in a given class. We tune the hyper-parameters on the validation set and report results in terms of top-1 accuracy averaged over the unseen classes. As shown in Table 5 our approach significantly outperforms other approaches.

## 5  Conclusions

We have proposed a novel clustering based regularization which encourages parsimonious representations, while being easy to optimize. We have demonstrated the effectiveness of our approach on a variety of tasks including unsupervised learning, classification, fine grained categorization, and zero-shot learning. In the future we plan to apply our approach to even larger networks, *e.g.*, residual nets, and develop a probabilistic formulation which provides a soft clustering.

**Acknowledgments**

This work was partially supported by ONR-N00014-14-1-0232, NVIDIA and the Intelligence Advanced Research Projects Activity (IARPA) via Department of Interior/Interior Business Center (DoI/IBC) contract number D16PC00003. The U.S. Government is authorized to reproduce and distribute reprints for Governmental purposes notwithstanding any copyright annotation thereon. Disclaimer: The views and conclusions contained herein are those of the authors and should not be interpreted as necessarily representing the official policies or endorsements, either expressed or implied, of IARPA, DoI/IBC, or the U.S. Government.

## Footnotes

[1] `https://github.com/lrjconan/deep_parsimonious`

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
