[Reviews · NeurIPS 2016]

Reviewer 1

Summary

This paper proposes to add a k-means style clustering term to the training objective function for deep networks. The clustering is enforced along multiple dimensions of a layer, though simple sample based clustering appears to be most effective. The paper treats these clustering terms as a regularizer and shows that it can sometimes be more effective than DropOut, L2-Weight decay etc.

Qualitative Assessment

This is generally a clearly written paper, with a simple and somewhat effective idea, and some nice visualization results that lend interpretability to the model. The main shortcoming of the paper is that its key contribution: that of training deep networks in the presence of a k-means clustering term is fairly underwhelming in terms of technical novelty when compared to typical NIPS papers. Answers or insights to the following questions may make the paper stronger: * When is a certain type of clustering more effective, e.g., sample clustering in Table 3, spatial clustering in Table 5? * Why not combine different types of clustering mechanisms to see if their regularization effect is complimentary? * Were the parameters of the baseline approaches in Table 2 carefully tuned? * How do the results of Table 5 look like for newer architectures (e.g., inception family)? * How do the results vary as a function of number of cluster centers, and lambda? * The method may be seen as a form of semi-supervised learning. So some analysis of the effectiveness of clustering as a function of increasing number of labeled samples would have been interesting.

Confidence in this Review

2-Confident (read it all; understood it all reasonably well)


Reviewer 2

Summary

The paper introduces a new regularization for deep networks, based on clustering activations over either samples, spatial dimensions or feature channels, in a 2-D convolutional network. The regularization is the k-means cost function computed in a given layer by performing one iteration of the K-means algorithm using cluster centers computed in previous iterations. The new cluster centers are also smoothed against the previous cluster centers, so that in effect there is a different learning rate for the K-means component. The experiments show improvements in generalization performance as a result of the regularization.

Qualitative Assessment

The overall method seems novel and interesting, and good results are obtained. However, the overall procedure makes a few approximations that raise the following important questions, which remain unanswered. 1) For the sake of training speed, the K-means objective is updated in an online way during optimization of the network. Thus it is not clear that the cluster centers and assignments that are used in the regularization term are properly synchronized to the neural network activity. Would the results be different if the clustering optimization were carried out to convergence in each epoch? Is it possible that the effect of regularization has more to do with the rate of convergence of the K-means algorithm than it has to do with the clustering itself, especially considering the benefits of early stopping? 2) The cluster centers and assignments are treated as constants in the regularization term, whereas they are actually functions of the network activations. What is the justification for holding them constant rather than unfolding the iterations of K-means (or soft K-means for the sake of differentiability)? Would the effect of the regularization be different if the derivatives were also passed through the steps of the clustering algorithm? --------- Thanks for the author feedback, which answers both points 1) and 2).

Confidence in this Review

3-Expert (read the paper in detail, know the area, quite certain of my opinion)


Reviewer 3

Summary

The paper suggests an approach to regularizing neural networks by encouraging their activations to form tight clusters in one of several proposed spaces. A k-means style objective is added to the overall network loss, penalizing deviation of activations in each layer from their closest (layer specific) cluster center. The cluster centers are learned jointly with the network weights. Three approaches are suggested for clustering: sample clustering, where the vectors to be clustered (at each layer) are the vectorized activations corresponding to different samples in a mini-batch; spatial clustering, where the vectors to be clustered are values across channels corresponding to different locations (and samples) within a convolutional layer's activations (same as sample clustering for fully connected layers); channel co-clustering, where vectors to be clustered correspond to vectorizations of different channels (and samples) within a convolutional layer's activations. The approaches are demonstrated to have a positive effect on generalization error on multiple datasets and across several tasks, including unsupervised learning on MNIST, classification on CIFAR10 and CIFAR100, and fine-grained classification and zero-shot learning on CUB-200-2011 (birds). Additionally, the clusters obtained provide insight into the structures learned by the network.

Qualitative Assessment

This is a very solid paper all around. The idea to cluster activations and force the network to "stay close" to a restricted set of representations is intuitive, acting as a parsimony constraint that additionally enables interpretability. While solid theoretical motivation and analysis isn't provided, the proposed algorithm feels natural and the experiments are fairly comprehensive and compelling. A broad range of tasks are considered, including unsupervised, fine-grained, and zero-shot learning in addition to standard classification, and the visualizations of cluster structure demonstrate that the cluster centers are meaningful---they convince me that the approach is performing as one would hope. Some specific points: It would be nice to see a table or chart demonstrating the effect of different choices of cluster size on generalization error (it is mentioned that cross-validation was used, but it would still be good to get a feel for the sensitivity with respect to choice of cluster size). A concern about the method is of course the added computation in each iteration required to update and utilize the clusters. It would be nice to get a sense of how much the proposed approach slows down training (e.g., report time per iteration with and without clustering). Finally, the description of how zero-shot clustering is performed is somewhat vague, and requires consulting the listed references. It would be nice to have a clearer description in the text.

Confidence in this Review

2-Confident (read it all; understood it all reasonably well)


Reviewer 4

Summary

This paper proposes a clustring-based regularization for training deep network. Three kinds of clustering-based regularization are provided. The weights of the network are learned with an online update. The experiments evaluate the approach in terms of generalization abilities and interpretablity of the learned representations.

Qualitative Assessment

It is not clear to me why clustering-based regularization facilitates genearlization and supports interpretability of the learned representations. Is there any theoretical guarantee or intuitive motivation? The paper is weakened without a clear motivation. Many regularization methods are introduced in the Related Work. The authors clearly illustrate the difference between the proposed approach and the related studies. However, there are few discussions about the advantages of the proposed algorithm over state-of-the-art. More discussions would be helpful. Also the exact meaning of 'parsimonious representation' is needed. The results in Talbe 1 are not impressive. How about the running time for these methods? In subsection 4.1, the number of clusters is set to 100. The actual number of clusters for MNIST is 10. Why set 100, not 10, in the experiments? A large difference in the value of \lambda is found in subsections 4.1 and 4.2, in which one is set to 0.0002, the other is set to 10. How to explain this large difference? The pages of the articles in the references are missing. Page 5, please provide the full name of SGD.

Confidence in this Review

2-Confident (read it all; understood it all reasonably well)


Reviewer 5

Summary

This paper proposed a clustering based regularization term for deep models. The term can be formulated by sample clustering, spatial clustering, and channel co-clustering. The objective function is optimized by an iteratively online update method.

Qualitative Assessment

This paper proposed a clustering based regularization term for deep models. The term can be formulated by sample clustering, spatial clustering, and channel co-clustering. The objective function is optimized by an iteratively online update method. However, it is not clear what is brought about by the proposed regularization term. What does the parsimonious representation mean and how the proposed method achieves the parsimonious representation? This paper lacks a theoretical justification for the proposed method. 1. Since the regularization term is based on clustering, how the number of clustering centers is set? I think this parameter may influence the performance a lot. Another parameter is the \lambda, I think it influence the performance a lot as well. This paper did not provide the determination of these parameters. I is not clear whether the proposed method is sensitive to these parameters. 2. The optimal condition of the regularization term is that all the representations and clustering centers have the same value because Y=f(X,W) where W is the network parameter set. When all the connecting weights equal to 0, the regularization term achieves the optimal value, i.e., 0. Then the regularization term is equivalent to reduce the discrimination ability. Therefore, the paper lacks theoretical analysis about the generalization of the proposed model which is not convincing.

Confidence in this Review

2-Confident (read it all; understood it all reasonably well)